


Published on behalf of

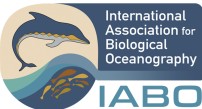

# eDNA metabarcoding shows highly diverse but distinct shallow, mid-water, and deep-water eukaryotic communities within a marine biodiversity hotspot

Patricia Cerrillo-Espinosa[1,2], Luis Eduardo Calderón-Aguilera[3], Pedro Medina-Rosas[1], Jaime Gómez-Gutiérrez[4], Héctor Reyes-Bonilla[5], Amílcar Leví Cupul-Magaña[6], Ollin Tezontli González Cuellar[7] and Adrian Munguia-Vega[2,8]

[1] Centro Universitario de la Costa, Universidad de Guadalajara, Puerto Vallarta, Jalisco, Mexico
[2] Applied Genomics Lab, La Paz, Baja California Sur, Mexico
[3] Departamento de Ecología Marina, Centro de Investigación Científica y Educación Superior de Ensenada, Ensenada, Baja California, Mexico
[4] Departamento de Ecología Marina, Centro Interdisciplinario de Ciencias Marinas, Instituto Politécnico Nacional, La Paz, Baja California Sur, Mexico
[5] Laboratorio de Sistemas Arrecifales, Universidad Autónoma de Baja California Sur, La Paz, Baja California Sur, Mexico
[6] Laboratorio de Ecología Marina, Centro Universitario de la Costa, Universidad de Guadalajara, Puerto Vallarta, Jalisco, Mexico
[7] Sociedad de Historia Natural Niparajá A.C., La Paz, Baja California Sur, Mexico
[8] Conservation Genetics Laboratory, School of Natural Resources and the Environment, University of Arizona, Tucson, AZ, United States of America

Corresponding author
Adrian Munguia-Vega, airdrian@arizona.edu

## ABSTRACT

As the impact of human activities continues to move beyond shallow coastal waters into deeper ocean layers, it is fundamental to describe how diverse and distinct the eukaryotic assemblages from the deep layers are compared to shallow ecosystems. Environmental DNA (eDNA) metabarcoding has emerged as a molecular tool that can overcome many logistical barriers in exploring remote deep ocean areas. We analyzed shallow water samples (<30 m) collected via SCUBA diving and adjacent deeper samples (mid-water 30–150 m, deep-water >200 m) obtained with Niskin samplers within 16 locations in a recognized hotspot of marine biodiversity (Gulf of California, Mexico). We sequenced an eDNA metabarcoding library targeting a fragment of the COI gene of eukaryotes. We demonstrated that the diversity of operational taxonomic units (OTUs) did not peak at shallow coastal regions and that the mid-water and deep-water benthic and pelagic samples had similar levels of biodiversity compared to shallow sites, but detected a significant vertical zonation between shallow and deeper habitats. Our results suggest that the deep refugia hypothesis, which posits that deep environments protect biodiversity during environmental changes, enabling species to survive and repopulate shallower regions, is not supported for most taxa and only applies to about a third (34.9%) of the 5,495 OTUs identified that were shared between the shallow and deeper layers. In comparison, the rest of the taxa were exclusive to either shallow (30.8%) or deeper zones (34.28%). The observation that mid-water and deep-water benthic and pelagic communities were as rich but quite distinct as shallow communities supports

extending spatial management and conservation tools to deeper habitats to include a significant fraction of unique phylogenetic and functional diversity.

## INTRODUCTION

As the footprint of human activities continues to expand in the tridimensional space of the oceans (*Halpern et al., 2019*), the attention to natural resources present in ocean depths also has increased beyond the shallow waters (<30 m), including the mid-water layer (30-150 m, also known as the mesophotic zone) and the deep-water ocean (>200 m depth) (*Puglise et al., 2009*; *Lauer & Reaka, 2022*). Interest in deeper regions of the oceans originate from abundant fish resources (*Irigoien et al., 2014*; *Pham et al., 2014*), a key role in nutrient regeneration and biochemical processes to sustain the ocean productivity throughout the biological pump (*Martin et al., 2020*), gas and oil exploration (*Cordes et al., 2016*), deep seabed mineral mining (*Jenner et al., 2023*), bioprospecting and biomimetics commercial applications (*Blasiak et al., 2022*) and the potential impacts of ocean-based climate interventions (*Levin et al., 2023*). The effective management and conservation of marine benthic and pelagic ecosystems and their ecological services require knowing their species distribution in time and space. Our understanding of marine biodiversity and the impact of human activities has historically focused on shallow coastal waters (<30 m) (*Webb, Vanden Berghe & O'Dor, 2010*; *Bongaerts et al., 2019*). However, mid-water and deep-water ecosystems have been considerably less studied (*Eyal et al., 2021*; *Jacquemont et al., 2024*). In some regions like the North Atlantic, the fauna from the mid-water layer is relatively well known from commercial fish monitoring (*Grimaldo et al., 2020*). Multiple logistical constraints explain the lag describing biodiversity beyond shallow waters, including the limited sampling accessibility, expertise and prohibitive costs of some of the most common exploration tools like deep-sea submersibles, trawls, remotely operated vehicles (ROVs) or sensors (*Bell et al., 2023*). One promising non-destructive technology that has boost exploration of biodiversity in deep ocean habitats is environmental DNA (eDNA) metabarcoding, where a sample of seawater or sediment is processed to obtain, amplify and sequence a conserved genomic region or barcode used to detect the presence and biodiversity of taxa within each sample (*Sinniger et al., 2016*; *Thomsen et al., 2016*). Portions of this text were previously published as part of a preprint (*Cerrillo-Espinosa et al., 2024*).

A key scientific question is how rich and distinct the benthic and pelagic biological communities from the mid-water and deep-water depths are compared to shallow ecosystems. A global meta-analysis suggested species diversity in the ocean decreases with depth, and that the 0–100 m depth range contains up to four times the diversity recorded between 100–200 m (*Costello & Chaudhary, 2017*). However, depending on the scale, other studies have shown species richness peaks in the mid-slope for fish (~200
m) and from 1,400 to 2,000 m across multiple taxa (*Piacenza et al., 2015*; *Saeedi et al., 2019*). The deep refugia hypothesis predicts that mesophotic (*i.e.,* mid-water depth) coral reefs may act as a refuge for the biota exposed to disturbances in shallow waters, and implicitly assumes most species display wide depth ranges and considerable vertical ecological connectivity between habitats (*Riegl & Piller, 2003*; *Bongaerts et al., 2010*). Although this mechanism could help avoiding the regional extinction of species (*Del Monte-Luna et al., 2023*), its relevance beyond coral reef ecosystems is controversial. Multiple observational studies in fish have shown that mid-water depth communities are diverse, but taxonomically and functionally distinct from their shallow counterparts (*Rocha et al., 2018*; *Medeiros et al., 2021*; *Loiseau et al., 2022*). The evidence supporting vertical connectivity within fish species occurring at different water column depths due to diel vertical migration is mixed (*Tenggardjaja, Bowen & Bernardi, 2015*; *Loya et al., 2016*). Studies on benthic communities have shown strong vertical zonation in function of seafloor depth, low connectivity, and a clear distinction between shallow and mid-water benthic communities (*Bongaerts et al., 2017*; *Stefanoudis et al., 2019*). However, traditional sampling methods and morphological identification of species may have taxonomic biases that can be addressed with eDNA to complement biodiversity knowledge gaps, for example, by including pelagic species with broad diel vertical migrations that are under-studied regarding the deep refugia hypothesis (*Lauer & Reaka, 2022*).

Our study focused in the Gulf of California (Fig. 1), a globally recognized hotspot of marine biodiversity on the Northwest region of Mexico (*Roberts et al., 2002*). The Gulf of California is ~1,500 km long, 40–241 km wide covering 12 degrees of north latitude and characterized by seasonally reversing ocean gyres that sit on deep basins reaching up to 4 km deep (*Munguia-Vega et al., 2018*). The Gulf of California is also a highly productive tropical-subtropical system that supports more than half of Mexico's marine fisheries and an economically profitable growing ecotourism industry. However, the Gulf of California shows signs of significant ecosystem decline due to overfishing and climatic change (*Gilly et al., 2022*). A recent eDNA metabarcoding study showed that biodiversity levels from shallow coastal areas in the Gulf of California are much higher than previously assumed based on historical data and visual surveys (*Mac Loughlin et al., 2024*), but few records exist on the biota from mid-water and deep-water ecosystems within the Gulf of California (*Morzaria-Luna et al., 2018*). The few studies available have focused on the central Gulf (Fig. 1A) using fish ROV surveys from the mid-water zone (*Hollarsmith et al., 2020*; *Velasco-Lozano et al., 2020*; *Velasco-Lozano et al., 2024*) and ROV and submersible fish and invertebrate surveys of the deep-waters (*Aburto-Oropeza et al., 2010*; *Portail et al., 2016*; *Gallo et al., 2020*). Reef-building coral ecosystems reach the limit of their northern distribution in the southern Gulf of California, where they are scarce and generally form poorly developed colonies (*Reyes & Lopez-Perez, 2009*). In contrast, the main habitats are represented by rocky reefs dominated <30 m depth by macroalgae (*e.g., Sargassum* sp.) and surrounded by rhodoliths and halophytes (seagrass beds, mangroves and saltmarshes) that transition to black coral forest (*Antipathes galapagensis)* between 30–200 m deep (*Munguia-Vega et al., 2018*; *Lavorato, Stranges & Reyes-Bonilla, 2021*). Although dozens of marine reserves have been established in the Gulf of California for conserving biodiversity

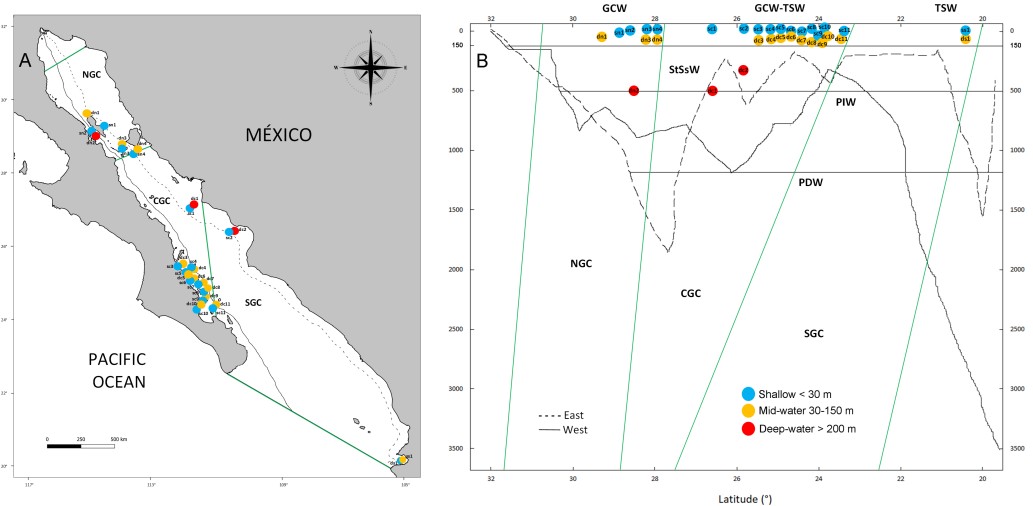

**Figure 1** **Geographic and bathymetric distribution of sampling sites in the Gulf of California.** Sampling sites distributed across the three biogeographic regions including the Northern (NGC), Central (CGC) and Southern (SGC) Gulf of California (A). Bathymetric profile of the Gulf of California, indicating sampling sites according to latitude (X axis) and depth (Y axis), sea water masses (horizontal lines) and biogeographic regions (green lines) (B). Shallow sites are in blue color, mid-water sites are in orange and deep-water sites are shown in red. Continuous and dashed lines in A and B show the location and depth profile of two transects in the western and eastern margins of the Gulf, respectively. See Table 1 for the classification of water mass for each sample.

and improving fisheries, they are focused on shallow coastal habitats (*Morzaria-Luna et al., 2018*; *Munguia-Vega et al., 2018*), while deeper benthic and pelagic habitats are not recognized as biodiversity hotspots.

We analyzed water samples from shallow sites (<30 m) collected *via* SCUBA diving, and deeper samples (mid-water 30–150 m or deep-water >200 m) collected during oceanographic cruises with Niskin samplers at the three biogeographic regions of the Gulf of California through eDNA metabarcoding a fragment of the COI gene targeting eukaryotes (Table 1). We tested if the deep refugia hypothesis applies to benthic and pelagic communities in a place with high levels of biodiversity and steep bathymetric profiles. Evidence on the diversity and distinctiveness of each vertical layer in the benthic and pelagic ecosystem could lead to a re-assessment of the ecological value and have cascading management and conservation implications. Our research goals were: (1) to contrast the levels of eukaryotic richness across shallow, mid-water and deep-water layers from the three biogeographic regions of the Gulf of California and (2) to establish the level of community structure and the proportion of shared taxa across vertical layers from surface to 500 m depth.

## MATERIALS & METHODS

### Sampling design

The Gulf of California is a marginal semi-closed sea of the north-eastern Pacific Ocean that has been recognized with three clearly defined biogeographic regions (North, Central

Cerrillo-Espinosa et al. (2025), *PeerJ*, DOI 10.7717/peerj.19249

**Table 1  Characteristics of the water samples collected for eDNA analyses.** For each of the 16 sampling sites, we provide details of the paired shallow, mid-water and deep-water samples, including biogeographic region, year and month of sampling, sampling site name, site ID, sample depth (SD), sample vertical distance to the bottom in the water column (SB), geographic distance between shallow and deep samples within each site (D), water temperature during sampling (T), corresponding water mass of the sample (WM) following *Lavín & Marinone (2003)*: GCW, Gulf of California Water; TSW, Tropical Surface Water; StSsW, Subtropical Subsurface Water; PIW, Pacific Intermediate Water.

| | | | | Shallow (<30 m) | | | | | Mid-water (30–150 m) and Deep-water (>200 m) | | | | | |
|---|---|---|---|---|---|---|---|---|---|---|---|---|---|---|
| Region | Year | Month | Site | ID | SD (m) | SB (m) | T(° C) | WM | ID | SD (m) | SB (m) | T(° C) | WM | D (km) |
| NGC | 2021 | July | Ángel de la Guarda | sn1 | 25 | 35 | 22 | GCW | dn1 | 40 | 4 | 18 | GCW | 45.65 |
| | 2018 | July | Canal de Ballenas | sn2 | 10 | 800 | 25 | GCW | dn2 | 500 | 310 | 12 | PIW | 0 |
| | 2021 | July | San Esteban | sn3 | 15 | 8 | 21 | GCW | dn3 | 80 | 7 | 15 | GCW | 2.95 |
| | 2021 | July | San Pedro Mártir | sn4 | 20 | 12 | 23 | GCW | dn4 | 110 | 83 | 16 | GCW | 2.79 |
| CGC | 2018 | July | Guaymas | sc1 | 10 | 1760 | 27 | GCW | dc1 | 500 | 1270 | 15 | PIW | 0 |
| | 2018 | July | Huatabampo | sc2 | 10 | 500 | 28 | TSW | dc2 | 300 | 210 | 16 | StSsW | 0 |
| | 2021 | October | San Damián | sc3 | 19 | 14 | 28 | TSW | dc3 | 108 | 7 | 16 | TSW | 1.42 |
| | 2021 | October | San Marcial | sc4 | 9 | 2 | 28 | TSW | dc4 | 78.6 | 71.4 | 21 | TSW | 2.01 |
| | 2021 | October | San Mateo | sc5 | 9 | 3 | 28 | TSW | dc5 | 109.4 | 11 | 15 | TSW | 16.52 |
| | 2021 | October | Punta Botella | sc6 | 18 | 5 | 28 | TSW | dc6 | 104 | 138 | 16 | TSW | 2.59 |
| | 2021 | October | Santa Cruz | sc7 | 22 | 12 | 27 | TSW | dc7 | 108 | 160 | 17 | TSW | 1.56 |
| | 2021 | October | Las Animas | sc8 | 20 | 21 | 28 | TSW | dc8 | 110 | 65 | 16 | TSW | 1.5 |
| | 2021 | October | San Francisquito | sc9 | 24 | 18 | 29 | TSW | dc9 | 108 | 96 | 15 | TSW | 3.18 |
| | 2021 | October | Punta Coyote | sc10 | 3 | 2 | 28 | TSW | dc10 | 105 | 4 | 16 | TSW | 6.03 |
| | 2021 | October | El Bajo | sc11 | 25 | 25 | 29 | TSW | dc11 | 50 | 13 | 22 | TSW | 0 |
| SGC | 2021 | May | Yelapa | ss1 | 20 | 325 | 21 | TSW | ds1 | 50 | 147 | 17 | TSW | 0 |

and South, Fig. 1A) (*Morzaria-Luna et al., 2018*). Due to its oceanic connection to the Eastern Tropical Pacific and its location between temperate and tropical biogeographic regions, it is influenced by at least six water masses, each characterized by specific ranges of salinity, temperature and depth (Fig. 1B) (*Lavín & Marinone, 2003*; *Monreal-Jiménez et al., 2021*). Pacific Deep Water (PDW) is distributed below 1,200 m depth; Pacific Intermediate Water (PIW) below 500 m depth; Subtropical Subsurface Water (StSsW) below 150 m; Tropical Surface Water (TSW) found at the surface at the Southern and Central Gulf of California; Gulf of California Water (GCW) found at the surface in the Central and Northern Gulf of California and formed by evaporation of StSsW and TSW; modified California Current Water (CCW) between the 0–150 m, which is present in small volumes only close to the mouth of the Gulf of California (*Lavín & Marinone, 2003*; *Monreal-Jiménez et al., 2021*). We collected water samples at 16 sites distributed along the three biogeographic regions of the Gulf of California under research permits from SAGARPA-Comisión Nacional de Acuacultura y Pesca de México PPF/DGOPA-035/20 and the University of Arizona IACUC 2021-0777. Within each site, the experimental design included a shallow water sample (<30 m, $n = 16$) collected during SCUBA diving with sterilized 1 L Nalgene bottles (6 L total collected at different times during the dive at each site), and a paired deeper sample from either the mid-water layer (30 to 150 m, $n = 13$) or deep-water layer (>200 m, $n = 3$) collected with a 6 L Niskin bottle operated by hand or attached to an oceanographic rosette (Fig. 1B, Table 1). Samples were collected aboard diving vessels (D/V Quino El Guardian) or oceanographic cruises (CAPEGOLCA, R/V El Puma), approximately around noon.

The average distance between paired shallow and mid/deep water sampling sites was 2.7 km (except for one site where samples were 45 km apart, Table 1). The vertical position of the water sample in the water column was examined by measuring the distance between the sampling depth to the bottom, following GEBCO bathymetry (*GEBCO Bathymetric Compilation Group, 2021*) (Table 1). At 12 sites, the shallow samples represented benthic ecosystems sampled <35 m above the bottom, while at four sites shallow samples were pelagic and sampled 325 m to 1760 m above the bottom (Table 1). At 10 sites, the mid and deep-water samples were associated with benthic ecosystems near the bottom (sampled on average 36 m above the bottom), while at six sites the deeper samples represent pelagic environments sampled between 138 m to 1,270 m above the bottom, Table 1.

We used a clean and dedicated area of the research vessel to filter the water to minimize DNA contamination. All the collecting and filtering equipment was cleaned by submersion in 1% sodium hypochlorite solution between sampling events and rinsed thoroughly with running freshwater. On every sampling day, 2 L of running fresh water from the vessel used for cleaning the sampling and filtering equipment was collected and filtered in the field as a field control to test for external contamination. For each of the 32 sites and depth combinations (Table 1), we had three replicate filters, through which 2 L of water were filtered. Water samples were filtered with an electric pressure pump and nitrocellulose Millipore filters with a pore size of 0.45 $\mu$m placed in a Millipore Sterifil

filter unit. Each filter was deposited in a 15 ml falcon tube with silica during field work (*Miya & Sado, 2019*) and refrigerated at 8 °C back in the lab until processing.

## DNA extraction

DNA from environmental samples and negative field controls were extracted from the nitrocellulose filters with the DNeasy Blood & Tissue kit (QIAGEN,) according to the manufacturer's instructions and using a QIAvac 24 Plus vacuum manifold to minimize contamination and handling. A blank negative control was incorporated at each extraction event. Total DNA concentration was measured for each sample with the Qubit 2.0 fluorometer (Thermo Fisher Scientific) and the High Sensitivity assay. All eDNA extractions were performed in a dedicated eDNA room and inside a hood used solely for this purpose. The hood and all the equipment and materials were sterilized with 1% sodium hypochlorite solution and UV light for 20 min between extraction events. Filter tips were used in all pipetting to reduce the risk of cross-contamination among water samples.

## Library preparation and sequencing

Sequences (313 bp) of the cytochrome oxidase subunit I (COI) barcode were amplified per triplicate for each individual DNA extraction (*i.e.*, nine PCR1 for each shallow and deep layer per site, respectively) with primers mICOIintF-XT: 5' GGWACWRGWT-GRACWITITAYCCYCC 3' (*Wangensteen et al., 2018*) and dgHCO2198: 5'TAIACYTCIG-GRTGICCRAARAAYCA 3' (*Geller et al., 2013*). These primer sets contained a standard Illumina adapter and an anchoring site for the PCR2 primers that contained dual unique 8 bp indexes, following a library design provided previously (*Valdivia-Carrillo et al., 2021*). The amplification protocol for a 12 µl volume reaction included: 5 µl of eDNA ($\geq$ 2 ng), nuclease-free water, PCR Buffer (1X), MgCl2 (2.5mM), dNTP's (0.2mM), Primers (0.4 µM each), 0.02% of BSA and 2 U of Platinum Taq HiFi polymerase (Invitrogen). PCR conditions were 95 °C for 5 min, followed by 35 cycles of denaturation at 95 °C for 30 s, annealing at 45 °C for 30 s, an extension of 72 °C for 30 s and a final extension of 72 °C for 5 min. PCR negative controls were included in each PCR. A simulated mock community constructed from equimolar concentrations of DNA from 25 known species of fish and invertebrates from six different phyla from the Gulf of California was used as a positive control and is described in detail previously (*Mac Loughlin et al., 2024*). Amplification of final products was verified in 1.2% agarose gels stained with GelRed (Biotium).

During the library preparation, the PCR2s were conducted in triplicate for all field samples, the mock community, three pools of field controls, three pools of DNA extraction controls, and three pools of PCR controls. The PCR2 protocol for a final volume of 12 µl was: 3 µl from PCR1 pool, nuclease-free water, PCR Buffer (1X), 2.5 mM of MgCl 2, 0.2 mM of dNTP's, 0.4 µM each primer, 0.02% BSA and 1 U Platinum Taq HiFi polymerase (Invitrogen). The PCR2 thermocycling protocol was as follows: 95 °C for 5 min, 12 cycles of 95 °C for 30 s, 60 °C for 30 s, 72 °C for 30 s, and a final extension of 72 °C for 5 min. The estimated size of the amplicon (448 bp) was verified in 1.2% agarose

gels. The PCR2 products of the three replicates for each sample were pooled in a single tube and cleaned with AmpureXP beads (1.8X) (Beckman Coulter). The final products were quantified with the Qubit 2.0 fluorometer, standardized to equimolar concentrations and pooled for sequencing. The high-throughput sequencing of the library was carried out on the Illumina MiSeq platform (250 bp × 2) at the University of Arizona Genetics Core.

## Sequence analysis

The bioinformatic analysis was performed in a Linux Ubuntu system v.20.04.1 (*Sobell, 2015*) using the USEARCH v11 software (*Edgar, 2010*). Raw demultiplexed sequence reads were merged by maximum (380 bp) and minimum (280 bp) lengths where short alignments (<16 bp) were discarded, along with forward and reverse primers. The reads quality filter was done under a maximum expected number of errors 1.0. The reads were dereplicated with a minimum size (2 reads) to get the unique sequences and subsequently clustered (97% similarity threshold) into operational taxonomic units (OTUs) using the UPARSE algorithm (*Edgar, 2013*), including detection and exclusion of chimeras. The last step consisted of the generation of the OTU table.

## Taxonomic assignments

Taxonomic assignments for each OTU were evaluated through the NCBI nucleotide database (*Benson et al., 2018*) (access date: March 19th, 2023) using the BLAST algorithm matching highly similar sequences. We generate XML files of the first one hundred results obtained for each OTU. The XML files were read in the MEGAN 6 Community Edition software (*Huson et al., 2016*) with parameters: Min score of 50.0, Min Percent Identity of 70.0, and Min Support Percent of 0.01. MEGAN used the Tree of Life from NCBI, the Last Common Ancestor algorithm (LCA, 100% to cover and the naive approach). Each OTU was statistically assigned to the LCA of the top 10% of the hits within the taxonomic tree, where the less consistency of taxonomic assignment, the higher up in the tree the assignment is placed for the OTU until the LCA of all likely assignments is reached. The taxonomic assignments were manually checked to discard cross-sample contamination and remove sequences of bacteria, terrestrial and freshwater taxa. OTUs with no hits and no assignments in NCBI were compared against the BOLD Systems platform (access date: April 14th, 2023) (*Ratnasingham & Hebert, 2007*) with the following similarity threshold: 100–97% (species), 97–94% (genus), 94–91% (family), 91–88% (order), 88–85% (class) and <85%–>70% (phylum) following *Valdivia-Carrillo et al. (2021)*. All the OTUs found in the negative controls were removed from the dataset.

## Statistical analysis

Histograms to explore the distribution of OTUs per sample were created with the ggplot2 package and Venn Diagrams with the Venn Diagram package in R (*R Core Team, 2018*) and RStudio v2022.02.0 (*R Studio Team, 2020*). OTU richness was estimated with Chao 1 non-parametric estimator based on the abundance of rare OTUs using the Primer v7 software (*Clarke & Gorley, 2015*). To analyze the OTU richness (alpha diversity) and test for significant differences between biogeographic regions and between depths, we used the

Kruskal-Wallis test (95% CI). The statistical analyses and the graphics were performed in XLSTAT software (Lumivero). We used the Jaccard presence/absence dissimilarity matrix and a 2D non-metric multidimensional scaling (nMDS) ordination analysis in Primer v7 to visualize community differences contrasting biographic region, water column depth, and water mass. A permutational analysis of variance (PERMANOVA) was conducted in the PERMANOVA+ package (Primer v7) to analyze the community structure (beta diversity). Datasets were transformed into a presence/absence matrix, and the Jaccard index was used to perform global PERMANOVA analyses for all biogeographic regions (North, Central, South), depths (shallow, mid-water, deep-water) and water masses (GCW, TSW, PIW, StSsW). Additionally, we performed pairwise PERMANOVAs for all permutations of each level of biogeographic region, water column depth, and seawater mass. All the analyses were performed with 9,999 permutations. The *P* values were adjusted following the Benjamini–Hochberg method for multiple tests with a false discovery rate 0.05 (*Benjamini & Hochberg, 1995*). Finally, a Spearman correlation rank analysis was performed in XLSTAT to assess the relationship between OTU richness and water column depth.

## RESULTS

The sequenced library resulted in 4,665,588 total paired reads for the 32 samples, including controls (data deposited in GenBank Bioproject ID PRJNA1073001), with an average of 144,706 raw reads per sample (excluding controls) (Table S1). The USEARCH pipeline removed 2,270,370 reads during the merge step and 159,703 through quality control. The clusters <2 sizes were discarded (298,412 reads), along with 747,243 singletons and 31,775 chimeras. An OTU table was constructed from 2,110,667 reads, resulting in 228,953 unique reads grouped into 11,922 OTUs. The negative controls resulted in a total of 1,586 reads (Table S2), represented mostly by bacteria and the phyla Apicomplexa, Amoebozoa, Arthropoda, Mollusca, Cnidaria, Rhodophyta, Bacillariophyta and OTUs with no taxonomic assignment. We discarded a total of 6,427 OTUs, including those assigned to bacteria (2,468), terrestrial taxa (444), with no hits or taxonomic assignments (3,229) and all OTUs found in the negative controls (188, Table S2). The final analyses were conducted with 5,495 OTUs (Table S3). From these, 4,493 were taxonomically assigned with BLAST on the NCBI database, and 903 were assigned with the BOLD Systems platform. A total of 1,694 of these OTUs (30.8% from the total) were either assigned above the phylum taxonomic rank within Eukaryotes, or these were taxa of the Stremenopiles, Alveolata, and Rhizaria lineages (SAR) for which higher level phylum taxonomy is still unresolved. The mock sample displayed a total of 4,065 reads grouped in 200 OTUs. Within the observed mock community, we successfully identified 20 taxa (80%) across various taxonomic levels. Out of the 25 expected species, we found a COI sequence in GenBank for only 15 species (60%) and detected two (8%) at the species level, two (8%) at the genus level, five (20%) at the family level, four (16%) at the order level, and seven (28%) at the class level. We observed wide variation in the number of OTUs and reads assigned to each taxa within the mock community (Table S4).

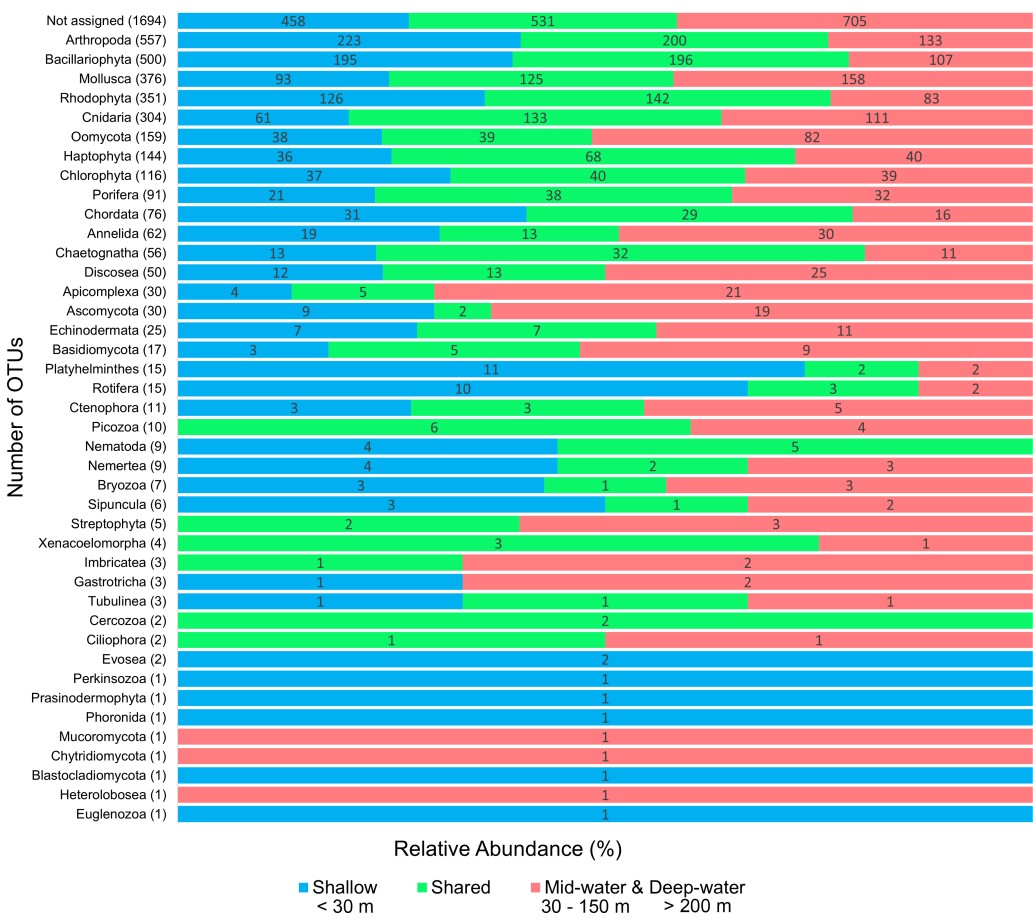

**Figure 2** **Number of eukaryotic OTUs found within 41 phyla in shallow and deep samples from the Gulf of California.** OTU counts for eukaryotes at the phylum level from seawater samples collected at shallow and deep layers (mid-water and deep-water) in the Gulf of California. Phyla are ordered showing decreasing OTU diversity from top to bottom. Numbers next to each phylum indicate the total number of OTUs, while numbers in each colored bar indicate the number and relative proportion of OTUs exclusive of shallow or deep samples, and those shared between depth layers.

We detected a total of 41 eukaryotic phyla among the samples analyzed. The five most important phyla in terms of number of OTUs were (in decreasing order): Arthropoda, Bacillariophyta, Mollusca, Rhodophyta and Cnidaria (Fig. 2). Most phyla showed a portion of OTUs that were unique to shallow or deep samples (combined mid-water and deep-water depths), and about one third of OTUs were found in both. Most phyla that were exclusive to shallow samples (Evosea, Euglenozoa, Blastocladiomycota, Phoronida, Perkinsozoa, and Prasinodermophyta), or exclusive to deep samples (Heterolobosea, Chytridiomycota, and Mucoromycota) were represented by a single OTU, respectively. Most of the taxonomically unassigned OTUs were exclusive to deep samples (705); followed by those OTUs shared between shallow and deep samples (531) and 458 OTUs were exclusive to shallow samples.

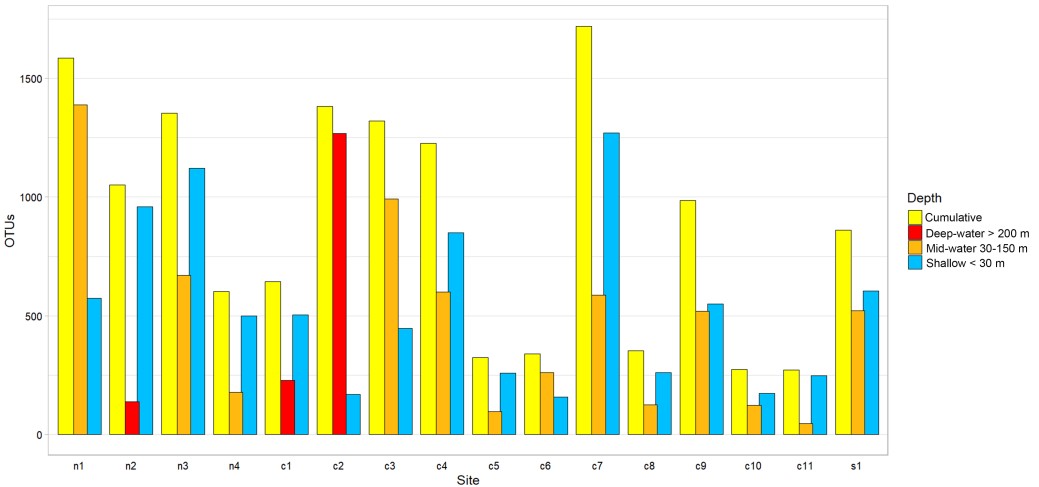

**Figure 3 Eukaryotic OTU richness found in 16 sampling sites from the Gulf of California.** OTU richness for each sampling site is ordered according to their latitudinal distribution (Northern samples on the left). Each site is represented by three vertical bars showing the number of OTUs found at shallow and deep (mid-water or deep-water) layers and the total or cumulative number of OTUs found.

The Chao1 species richness estimator did not reach the asymptote for the total number of samples, particularly for the deep samples (mid-water and deep-water depths), suggesting that the sampling effort was still insufficient to reach saturation of the species present in these benthic and pelagic communities (Fig. S1). The combined OTU richness from shallow and deep samples within sites averaged 893.4 OTUs (range = 272–1,719) and was highly heterogeneous among the 16 sampling sites and within the three layers compared across sites (Fig. 3). At a few sites, mid-water and deep-water samples showed more than double the OTU richness than their shallow counterparts (*e.g.*, n1, c2, c3, and c6) and some shallow sites had the opposite trend (*e.g.*, n2, n4, c1, c5, c7, and c11). The cumulative OTU richness observed allowed to estimate how many species are shared between the shallow and deeper layers within each site, *i.e.,* if all species are completely shared between depths, then the cumulative number would not be larger than any of the two estimates from each depth layer. The fact that the cumulative richness was always larger than the individual observations for shallow and deeper samples indicates that a fraction of the OTUs within each site were not shared between depth layers. The mean OTU richness of the shallow samples was higher than the mid-water layer, but similar to the richness of the deep-water layer (Shallow = 537.4 OTUs; Mid-water = 467; Deep-water depth = 541.6, Fig. 4A). Differences were not statistically significant between any of the depth layers pairwise comparisons (Kruskal-Wallis $p \geq 0.514$, Table S5). The mean OTU richness of the sites in the Northern Gulf of California (North = 688.2 OTUs) was comparatively higher than those collected in the Central (Central = 439.5) and Southern (South = 559.5) Gulf of California (Fig. 4B), but differences between the biogeographic regions pairwise comparisons were not significant (Kruskal-Wallis $p \geq 0.132$, Table S6).

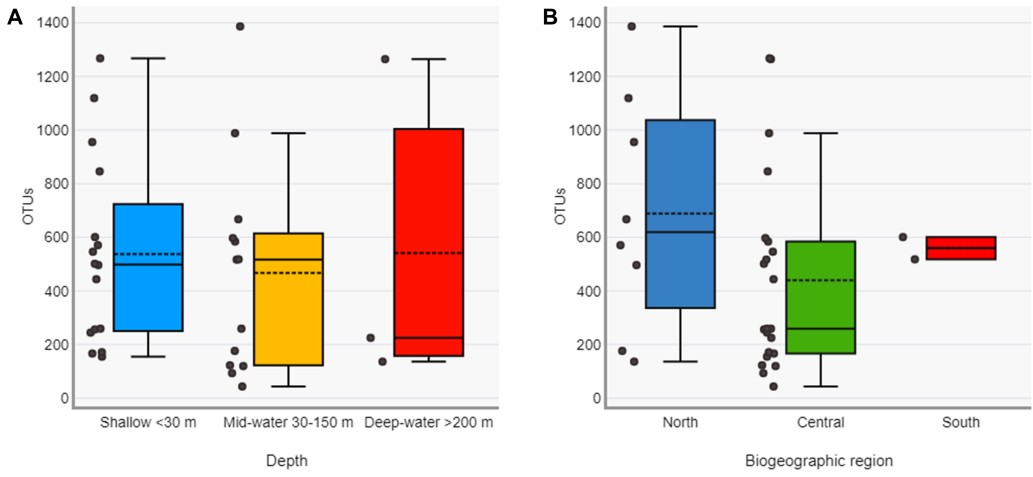

**Figure 4 Diversity of eukaryotic OTUs by depth layer and biogeographic region from the Gulf of California.** Comparison of OTU richness among shallow, mid-water and deep-water samples (A) and among North, Central and South biogeographic regions (B). Box plots showing the mean (dotted line), median (solid line), upper and lower quartiles and 1.5X de interquartile range (whiskers).

About one-third of the OTUs recorded in all sites from the Gulf of California were exclusive to shallow samples (30.8%), a third were exclusive to the deep samples (mid-water and deep-water layers) (34.2%), and a third were shared between both (34.9%, Fig. 5A). The analyses of taxonomic assignments at class level showed that most of classes were shared between shallow and deep samples (mid-water and deep-water). However, 16 taxa from 12 different phyla (Ciliophora, Pseudofungi, Amoebozoa, Platyherlminthes, Euglenozoa, Chordata, Bryozoa, Mollusca, Blastocladiomycota, Ascomycota, Basidiomycota and Chlorophyta) were exclusive to shallow water samples and nine taxa from eight distinct phyla (Choanozoa, Mucoromycota, Platyhelminthes, Charophyta, Ascomycota, Chytridiomycota, Chlorophyta and Bryophyta) were exclusive to mid-water and deep-water (Fig. 5B).

The stress value in the nMDS plots (0.14) reflected a fair quality fit ordination of the sampling sites. The first nMDS plot among biogeographic regions showed a concentration of sampling sites from the Northern Gulf (Fig. 6A). The eukaryotic community structure based on the presence-absence of species was significantly different across the three biogeographic regions (PERMANOVA $df = 29$, Pseudo-F = 1.6467, adjusted $p = 0.003$, Table S7) and between the pairwise comparisons North-Central (PERMANOVA $df = 28$, $t = 1.3235$, adjusted $p = 0.006$) and Central-South (PERMANOVA $df = 22$, $t = 1.2586$, adjusted $p = 0.013$), but not North-South (PERMANOVA $df = 8$, $t = 1.2196$, adjusted $p = 0.07$) (Fig. 6A). The second nMDS plot showed a separation of shallow, mid-water and deep-water layers (Fig. 6B), where the shallow water samples showed variability in the first axis but not in the second one. In contrast, samples from mid-water and deep-water layers were more heterogeneous. We also found significant differences across the three depths (PERMANOVA $df = 29$, Pseudo-F = 1.3647, adjusted $p = 0.015$) (Fig. 6B), although, after the pairwise comparison, we only found significant differences in the eukaryotic

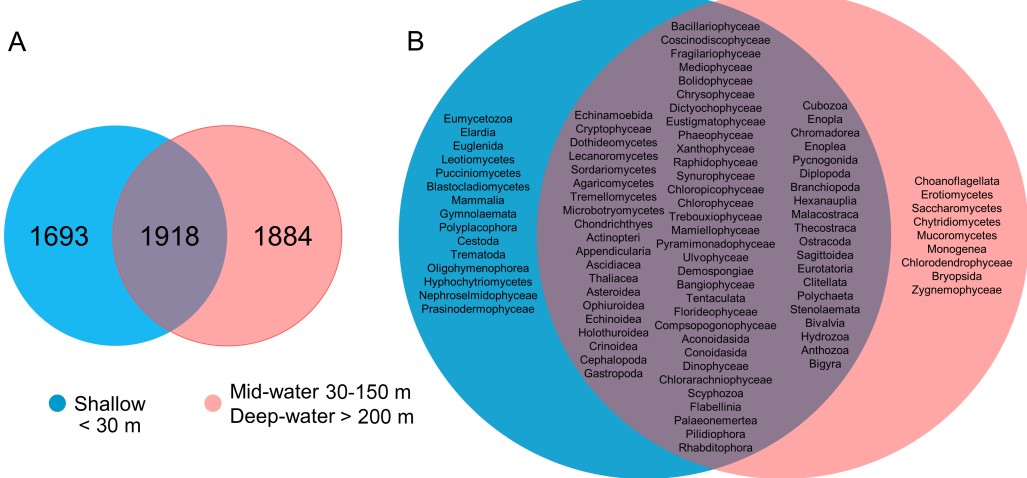

**Figure 5 Venn diagrams comparing number of eukaryotic OTUs and taxonomic classes by depth layer.** Venn diagram representing the number of exclusive and shared OTUs found in shallow and deep (mid-water and deep-water) samples (A), and the presence of exclusive and shared identified taxonomic classes (B).

communities between shallow *vs.* mid-water (PERMANOVA $df = 27$, $t = 1.3503$, adjusted $p = 0.008$). In the nMDS plot showing the association with water masses, the Gulf of California Water (GCW), represented by the shallow and mid-water sites from the North region and one from the Central region (sc1), formed a distinctive cluster (Fig. 6C). The Tropical Surface Water (TSW), represented by most of the rest of shallow and mid-water sites, were highly heterogenous. The communities of OTUs collected in the four different water masses also exhibited a significant difference in structure (PERMANOVA $df = 28$, Pseudo-F = 1.2656, adjusted $p = 0.015$). However, following the pairwise comparison, we found significant differences in OTUs recorded from water samples collected in the water masses: GCW-TSW (PERMANOVA $df = 27$, $t = 1.3067$, adjusted $p = 0.008$) and GCW-PIW: (PERMANOVA $df = 8$, $t = 1.2492$, adjusted $p = 0.045$) (Fig. 6C, Table S7). Finally, the Spearman rank correlation test suggested a poor relationship between water column depth and OTUs richness (R = −0.161), and the coefficient test result was not significant ($R^2 = 0.26$, $p = 0.377$) (Fig. 7, Table S8).

## DISCUSSION

We demonstrated that the benthic and pelagic eukaryotic biota at the mid-water (30-150 m) and deep-water layers (>200 m) showed similar levels of OTU richness compared to adjacent samples from the shallow layer (<30 m) at the same sites. We also found evidence supporting the view that the eukaryotic communities found in the deeper layers (mid-water & deep-water depths) were distinct from the shallow layer counterparts, where about a third of all the OTUs were exclusive to the deeper samples. Our results from the Gulf of California between the shallow, mid-water and deep-water layers do not support the observed trend of decreasing marine biodiversity with depth previously

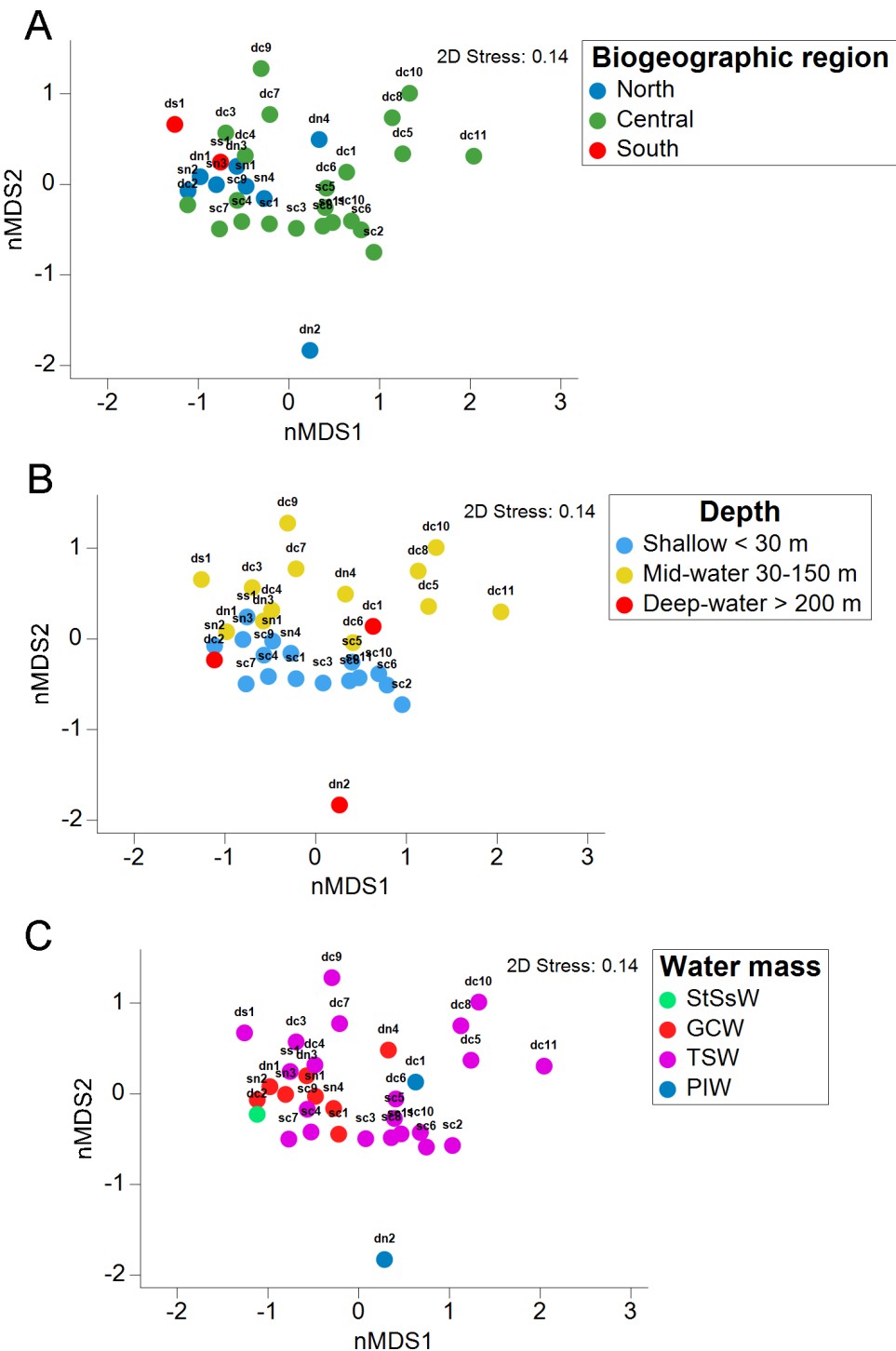

**Figure 6** Eukaryotic community structure by depth layer, biogeographic region and seawater mass from the Gulf of California. Non-metric multidimensional scaling (nMDS) ordination analysis based on the presence/absence data showing the OTUs community structure between three biogeographic regions in the Gulf of California (A), three water sampling depths (B), and four water masses (C).

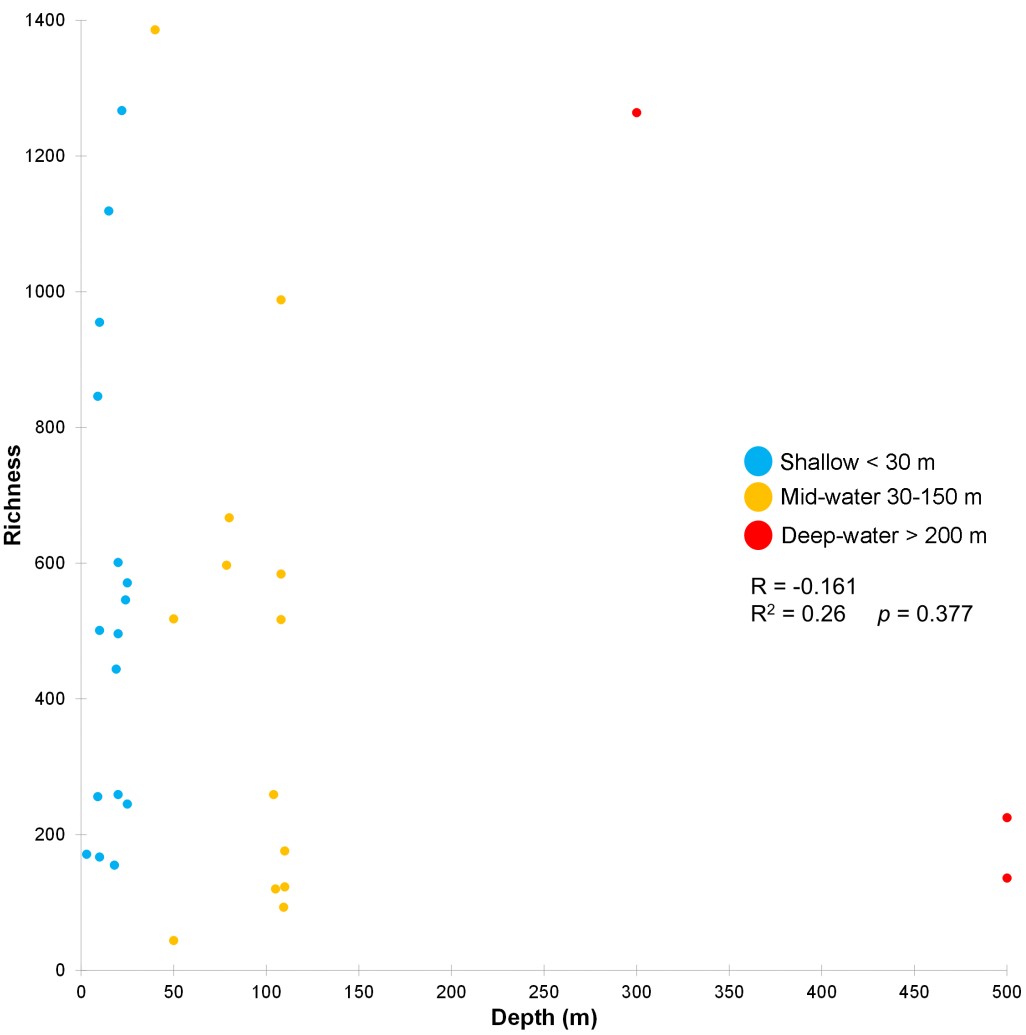

**Figure 7** **Relationship between depth and OTU richness for each sample.** Scatter plot showing the richness of OTUs (Y axis) as a function of sampling depth (X axis), including Spearman's rank correlation (R) and coefficient. Data points are colored according to the three depth layers.

reported (*Costello & Chaudhary, 2017*). Our observations are aligned with past studies of biodiversity trends from trawl surveys (*Piacenza et al., 2015*) and photo transects (*Stefanoudis et al., 2019*) along similar vertical depth gradients concluding that biodiversity of the mid-water and deep-water layers is richer and considerably distinct than previously assumed. Other reports employing eDNA metabarcoding using general biodiversity primers also have detected vertical changes in the community composition of eukaryotes comparable to the present study (*Zhang et al., 2020*; *Govindarajan et al., 2021*; *Cote et al., 2023*; *Hoban, Bunce & Bowen, 2023*). Our results imply that the deep refugia hypothesis could apply to only a third of all taxa identified that shared benthic and pelagic habitats between shallow and deeper layers (mid-water & deep-water depth), while two thirds of total OTUs could be considered exclusive to either shallow or deeper layers.

The vertical transition in coral reef biotic communities along a depth gradient has been explained by changes in temperature and availability of light and nutrients between the upper (30–60 m) and lower mid-water zones (60–150 m) (*Lesser et al., 2019*; *Slattery et al., 2024*). In coral reefs, this vertical change is characterized by a sharp change in foundational species from photoautotrophic (hard corals and macroalgae) to heterotrophic taxa (sponges and azooxanthellate octocorals) feeding on large (>2 mm) zooplankton that are more abundant in the mid-water layer (*Andradi-Brown et al., 2017*; *Lesser et al., 2019*). Fish species assemblages in the mid-water layer also show lower abundance of herbivores and higher population biomass (*Stefanoudis et al., 2019*; *Rocha et al., 2018*). Additionally, coral reef ecosystems show a vertical gradual transition to the deeper zone represented by rariphotic ecosystems with unique species assemblages located between 150–300 m that are also different from the mid-water biota (*Baldwin, Tornabene & Robertson, 2018*). Despite reef-building coral reefs not being dominant throughout the Gulf of California, a similar transition from shallow photoautotrophic (macroalgae and halophytes) to mid and deep-water heterotrophic taxa could explain the presence of different biotic communities. In the Gulf of California, there is a vertical transition below 30 m depth to Antipatharian black coral forests (*Antipathes galapagensis* and *Myriopathes panamensis*) and seafan reefs (*Leptogorgia*, *Eugorgia*, *Pacifigorgia*, *Muricea* and sea pen *Cavernulina*) that feed on abundant zooplankton (*Aburto-Oropeza et al., 2010*; *Lavorato, Stranges & Reyes-Bonilla, 2021*). For example, in the Gulf of California copepods do not migrate daily maintaining maximum densities at 50 m depth independently of the time in the circadian cycle (*Palomares-García, Gómez-Gutiérrez & Robinson, 2013*). Mesophotic rhodolith beds and algal communities have also been described in the mid-water layer (*Hollarsmith et al., 2020*). Below the 200 m some dominant species include glass sponges (Hexactinellida), and octocorals (*Anthomastus*, *Paragorgia*), along with deep-sea chemosynthetic ecosystems dominated by symbiont-bearing bivalves, polychaetes and other annelids (*Aburto-Oropeza et al., 2010*; *Portail et al., 2016*). Our eDNA analyses detected some of these habitat-forming taxa at higher taxonomic levels in multiple mid and deep-water samples, including octocorals, coralline red algae, and multiple unidentified sponges not found in the shallow layers (Table S3).

Marine biodiversity assessments have traditionally considered the ocean on a two-dimension scale with little focus on depth (*Jacquemont et al., 2024*). The advent of techniques like eDNA metabarcoding from water samples collected remotely (Niskin samplers or devices for water filtering *in-situ*) (*Hendricks et al., 2023*) opens new possibilities for characterizing marine biodiversity from deep ecosystems that were previously logistically inaccessible and thus data limited. The finding that mid-water and deep-water benthic and pelagic communities are as diverse as shallow water assemblages in the ecosystem of the Gulf of California and yet distinct in their species composition has some important implications for management and conservation. Based on a principle of complementarity that explains the gain in biodiversity representation upon the inclusion of a new area within an establish network of protected zones, our study supports the view that marine reserves and other effective area-based conservation measures should extend to include the biodiversity present in the mid-water and deep-water layers to maximize
protection of taxonomic and functional diversity of marine ecosystems against human impacts (*Robison, 2009*; *Loiseau et al., 2022*; *Jacquemont et al., 2024*). Thus, extending the bathymetric range on which marine reserves are placed and providing protection for the entire water column would benefit a third of all the benthic and pelagic taxa that were not present in the shallow habitats and were exclusive to the deeper layers of the Gulf of California. Recent studies suggest that some of the species that exclusively inhabit the deeper oceanic layers are threatened with regional extinction, but many of these taxa are data deficient (*Finucci et al., 2024*). Available information from other studies supports a strong ecological and biogeochemical connectivity between the pelagic (water column) and benthic environments, but more evidence needs to be collected regarding the influence of management tools and policies directed to different depth layers, particularly on oceanic marine reserves (*O'Leary & Roberts, 2018*; *Blanluet et al., 2024*).

Our study showed that, in the case of disturbances to shallow habitats, the deeper habitats could potentially contribute to the resilience and persistence of only about a third of the total benthic and pelagic taxa that was shared between shallow and deeper layers. This is similar to 36% of benthic species shared between the shallow (<30 m) and mid-water habitats (30–150 m) from a study of depth distributions for thousands of species in the Gulf of Mexico, where the number of shared pelagic species was almost twice (67%) (*Lauer & Reaka, 2022*). Some species are present between the shallow and deeper zones likely due to diel vertical migration (*Canals et al., 2021*). For example, krill migrates in the Gulf of California from surface to the mid-water layer every day explaining sharing of species in different vertical habitats (*Ambriz-Arreola et al., 2017*). However, the ecological and genetic connectivity could be limited for some species that do not migrate across distinct vertical ecosystem boundaries (*Palomares-García, Gómez-Gutiérrez & Robinson, 2013*; *Loya et al., 2016*; *Bongaerts et al., 2017*).

Some evidence suggest that ecosystems from the mid-water zone could act as a refugia for some key or foundational species (and the benthic communities associated with them) when extreme climatic events impact shallow habitats (*Giraldo-Ospina, Kendrick & Hovey, 2020*). Marine species track isotherms due their metabolic rates in both pelagic and benthic habitats. However, those isotherms can shift with climate change modifying the species distributions in time and space (*Lenoir et al., 2020*; *Pinsky, Selden & Kitchel, 2020*). This reorganization of marine biodiversity highlights the need to monitor climate-driven community restructuring *in-situ* at different ocean depths. Projected range shifts based on climate change rates are faster in the deep ocean compared to the near surface, particularly for the mid-water layer biota (*Brito-Morales et al., 2020*). In Australia, mid-water reefs are ecologically relevant habitat for several economically profitable commercial species (*Williams et al., 2019*) and the same pattern is true in the Gulf of California (*Velasco-Lozano et al., 2024*). Thus, the redistribution of species, both away and into mid-water habitats, will have economic impacts to the fishing sector (*Gordo-Vilaseca et al., 2023*; *McClure et al., 2023*) as has been documented already in Mexico (*Gilly et al., 2022*). With the availability of marine biodiversity data from the deep ocean, three-dimensional spatial prioritization analyses could be conducted and likely will become

more common in the near future (*Venegas-Li et al., 2018*), and could incorporate climate velocities to identify climate refugia within the present and future marine reserves (*Brito-Morales et al., 2022*).

While vertical stratification and biogeographic regionalization impacted eukaryotic community composition, we also found some evidence suggesting a role of water masses in the Gulf of California on species assemblage composition. Other studies have also shown distinct communities detected *via* eDNA metabarcoding associated to the different water masses and driven by different planktonic organisms (*Adams et al., 2023*). The vertical distribution of multiple water masses layered in long, narrow, and deep layers of the Gulf of California seem to contribute to a higher similarity of OTUs among sites located at similar depths and influenced by the same water mass. Each water mass represents a different habitat, characterized by multiple taxa responding to common environmental conditions (*Lima-Mendez et al., 2015*); in this case, the physical and chemical signatures that define it. If taxa are associated with water masses and the water masses change position, then the proportion of species in deep water refugia could also change. The Northern Gulf of California is characterized by strong tidal currents and complex topography that promotes year-round vertical mixing and primary productivity and high zooplankton biomass throughout the year (*Salas-de León et al., 2011*). The large difference observed between the community from the mid-water layer in the Northern Gulf of California (represented by sample dn2) from the rest of the samples separates the PIW water mass into two distinct regions previously reported in zooplankton (*Brinton, Fleminger & Siegel-Causey, 1986*; *Quiroz-Martínez et al., 2023*). This could be attributed to the oceanographic isolation of the Northern Gulf of California by an Archipelago with narrow channels and sills from the central Gulf of California (Fig. 1). However, we caution that our sample size for deep-water samples is small, and that eDNA results can show considerable variability in composition (*Stat et al., 2017*; *Cote et al., 2023*), suggesting further sampling efforts could help corroborate these patterns.

The use of eDNA metabarcoding for biodiversity monitoring beyond shallow coastal zones has several methodological challenges (*Hansen et al., 2018*; *Cote et al., 2023*; *He et al., 2023*). While modeling and empirical studies have shown that the vertical distribution of eDNA often corresponds to the vertical location of the organismal source (*Canals et al., 2021*), sinking of eDNA has been proposed and implies that eDNA could be detected in the upper depth limit of any given taxa (*Hansen et al., 2018*; *Allan et al., 2021*). Additionally, there is a possibility that our approach overestimates the proportion of taxa shared between the shallow and deeper layers due to the mixing of water masses combined with eDNA transport (*Hansen et al., 2018*). This should be analyzed considering the role of natural diel vertical migrations on plankton and nekton that impact eDNA distribution in the epipelagic and mesopelagic habitats (*Easson et al., 2020*; *Canals et al., 2021*). A decline of eDNA concentration as a function of depth and the presence of false negatives (*Cote et al., 2023*; *He et al., 2023*) highlights the need of larger water volumes and higher number of replicates to improve the detection of eukaryotic biodiversity from deep samples (*McClenaghan et al., 2020*; *Govindarajan et al., 2022*). Additionally, each metabarcoding marker is associated with taxonomic biases, and using multiple markers is recommended

to improve the taxonomic coverage of eDNA samples (*McClenaghan et al., 2020*; *Cote et al., 2023*). The proportion of taxonomically unassigned taxa of the present study indicates a poor taxonomic coverage from the deeper layers of the ocean in public genetic reference databases that has been highlighted previously (*e.g.*, *Duhamet et al., 2023*). The species accumulation curves suggest still incomplete sampling of communities and the need for larger sampling efforts including broader geographic and bathymetric coverage, and sequencing depth.

## CONCLUSIONS

Based on the findings of the present study, the eukaryotic communities of shallow, mid-water and deep-water samples of 32 sites collected across three biogeographic regions of the Gulf of California were characterized. Environmental DNA metabarcoding of the cytochrome c oxidase subunit I (COI) gene showed evidence of an unprecedent high biodiversity with a total of 5,495 operational taxonomic units (OTUs) including a wide range from species-level taxa to many without a current taxonomic classification. This underscores the necessity for enhanced sampling efforts, encompassing a broader spatial and vertical range resolution. Contrary to our initial hypothesis of decreasing biodiversity in function of increasing water column depth, the present study revealed that shallow water samples exhibited a eukaryotic diversity comparable to that of deeper water samples, with nearly equivalent numbers of exclusive species. This evidence did not support the deep refuge hypothesis for most of the benthic and pelagic taxa from the Gulf of California, suggesting that mid-water and deep-water pelagic layers could serve as refugia for only a third of the taxa that are shared between shallow and deeper habitats. We also demonstrated that vertical oceanographic gradients significantly influenced the taxonomic composition of eukaryotic communities. This study marks a pioneering research effort of eukaryotic biodiversity in the Gulf of California, analyzing water samples from mid-water and deep-water habitats, although analysis of additional deep-water samples is needed to corroborate some of the observed trends. The molecular evidence produced by the present research has the potential to inform improved resource management practices and the protection of deeper marine environments of the Gulf of California considering that approximately one-third of all the species detected were exclusive to the mid-water and deep-water habitats and not detected in the shallow samples.

## ACKNOWLEDGEMENTS

We thank Dr. Carlos Robinson for allowing us to collect samples during the 2018 CAPE-GOLCA cruise. We thank the logistical support from the crew of the diving vessel Quino El Guardian. We thank Alma Paola Rodríguez-Troncoso and Camila Mac Loughlin for help filtering water samples.

### Funding

The present study was funded by Consejo Nacional de Humanidades, Ciencia y Tecnología (CONAHCYT) de México, via the project: Challenging the deep refugia hypothesis and its implications under a climatic change scenario (CONAHCYT-CF2019-39210). Additional support was provided by The University of Arizona—CONAHCYT Binational Consortium for the Regional Scientific Development and Innovation (CAZMEX) to Adrian Munguia-Vega and Héctor Reyes-Bonilla, and National Geographic Society NGS-62186R-19 to Adrian Munguia-Vega. Deep water samples collected during R/V El Puma CAPEGOLCA oceanographic cruise were obtained with financial support of SIP20180084, SIP-2018-RE/021, CONACHYT Ciencia Básica CB-2016-01-284201, and UNAM-PAPIIT IN200666610-3 research grants. This paper is part of the PhD research program of Patricia Cerrillo-Espinosa in Biosistemática, Ecología y Manejo de Recursos Naturales y Agrícolas (BEMARENA) at the University of Guadalajara. Patricia Cerrillo-Espinosa is also the recipient of a PhD scholarship granted by CONAHCYT (CVU 268703). The funders had no role in study design, data collection and analysis, decision to publish, or preparation of the manuscript.

### Grant Disclosures

The following grant information was disclosed by the authors:
Consejo Nacional de Humanidades, Ciencia y Tecnología (CONAHCYT) de México: CONAHCYT-CF2019-39210.
The University of Arizona—CONAHCYT Binational Consortium.
National Geographic Society: NGS-62186R-19.
CONACHYT Ciencia Básica: CB-2016-01-284201.
UNAM-PAPIIT: IN200666610-3.
CONAHCYT: CVU 268703.

### Competing Interests

The authors declare there are no competing interests.

### Author Contributions

- Patricia Cerrillo-Espinosa performed the experiments, analyzed the data, prepared figures and/or tables, authored or reviewed drafts of the article, and approved the final draft.
- Luis Eduardo Calderón-Aguilera conceived and designed the experiments, performed the experiments, authored or reviewed drafts of the article, and approved the final draft.
- Pedro Medina-Rosas conceived and designed the experiments, authored or reviewed drafts of the article, and approved the final draft.
- Jaime Gómez-Gutiérrez performed the experiments, authored or reviewed drafts of the article, and approved the final draft.

- Héctor Reyes-Bonilla conceived and designed the experiments, authored or reviewed drafts of the article, and approved the final draft.
- Amílcar Leví Cupul-Magaña performed the experiments, authored or reviewed drafts of the article, and approved the final draft.
- Ollin Tezontli González Cuellar performed the experiments, authored or reviewed drafts of the article, and approved the final draft.
- Adrian Munguia-Vega conceived and designed the experiments, performed the experiments, analyzed the data, prepared figures and/or tables, authored or reviewed drafts of the article, and approved the final draft.

## Animal Ethics

The following information was supplied relating to ethical approvals (i.e., approving body and any reference numbers):

Institutional Animal Care and Use Committee, University of Arizona. IACUC Protocol 2021-0777.

## Field Study Permissions

The following information was supplied relating to field study approvals (i.e., approving body and any reference numbers):

Secretaria de Agricultura Ganaderia Desarrollo Rural Pesca y Almentacion (SAGARPA), Comision Nacional de Acuacultura y Pesca de Mexico, PPF/DGOPA-035/20.

## Data Availability

The raw short DNA sequence reads are available at GenBank Bioproject: PRJNA1073001.

## Supplemental Information

Supplemental information for this article can be found online at http://dx.doi.org/10.7717/peerj.19249#supplemental-information.

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
