# Peer review of "eDNA metabarcoding shows highly diverse but distinct shallow, mid-water, and deep-water eukaryotic communities within a marine biodiversity hotspot"

_PeerJ, doi:10.7717/peerj.19249_

## Round 0.1 · original submission · Major Revisions

Dear Authors

Two referees have now commented on your manuscript. Both of them evidenced the quality of the work but they have several concerns that should be addressed before publication. They did a great job in commenting the paper and I agree with them that major revisions are needed to improve the manuscript.

Reviewer 1 ·

Basic reporting

Overall the paper is well-written and well-reported, but there are some minor wording issues throughout that can occasionally make it difficult to follow. In general, I think there is some room for improvement in terms of the overall conclusions of the study.

A few comments are below, and more can be found in the attached annotated PDF.

Lines 342–3498: I think the discussion of these results needs to be reworded, because I had difficulty understanding how the conclusions were made based on the figure and the statistical results. For instance, there were separate ANOSIM tests for water mass, depth zone, and region, yet in the results the authors claim “the Tropical Surface Water (TSW) it is the most influential between shallow and mid-water depth sites, and where sampling sites grouped together independently of the region. The Gulf of California Water (GCW) basically influence the shallow and mid-water depth sites from the North region and one from the central region (sc1). The Pacific intermediate Water (PIW) and the Subtropical Subsurface Water (StSsW) slightly separate the mid-water depth samples from the rest”. I don’t understand where these interactions are coming from (e.g., the influence of water mass on depth separation) when each factor was analyzed separately.

Lines 367–378: It’s not clear to me how the three-way SIMPER analysis was performed. As I understand it, PRIMER allows for one-way and two-way SIMPER analyses. And I don’t entirely understand what it means to say something like “The difference between the North and Central latitudinal regions was mainly due to contribution of taxa associated with the deep zone”.

Throughout the text, the term “column water” is used. It should be “water column”. A search in the PDF recovered instances at lines 151, 257, 259, 263, 266, 270, 274, 277, 368, and 379, although there could be more.

The authors use the term “exclusive from” when I think they actually mean to say “exclusive to.” There was one case in the abstract, and also at lines 43, 327, 332, 387, 399, 438, and 439, although there may be others.

The names of taxonomic ranks (except specific epithet) should be capitalized (e.g., Arthropoda) but the ranks themselves don’t need to be (e.g., “phylum” and “class” rather than “Phyum” or “Class”).

It feels a little uncertain as to what is considered “deep”. There are some samples from 40 m that are listed as “deep” (in Figure 3), but I would definitely not consider 40 m to be deep, especially in comparison to 500 m.

I think the journal standard is to have uppercase letters for subfigure headings. Make sure this is consistent. For example, line 321 cites “Fig. 4a” and figures 4–6 use lowercase letters on the figures themselves.

In figure 5, I suggest you use more distinct colors for shallow and deep. The two chosen are very similar. Also, as before, what specifically constitutes shallow and deep? Similarly, figure 6 should have more distinct colors for the different categories. It’s difficult to tell apart similar shades of grey and between orange and red.

Experimental design

The experimental design is sound

Validity of the findings

The findings over appear valid and are believable.

I have some issues with the authors' invocation of the deep reef refugia hypothesis in the context of this study, since that hypothesis was developed for mesophotic coral reef ecosystems, and particularly for adjacent habitat. In the case of this study, it's unclear that the shallow/deep samples are adjacent or paired, and differences found might just be a result of different habitat.

Additional comments

Additional comments and suggestions can be found in the attached annotated PDF. Orange notes and red strikethroughs are suggestions for changes, yellow notes are comments, and magenta notes are questions. Highlighted sections have comments that should pop up if you hover over them in acrobat or adobe reader.

Annotated reviews are not available for download in order to protect the identity of reviewers who chose to remain anonymous.

·

Basic reporting

The methods need some clarification to understand the study. More detailed comments below.

Experimental design

Please see above comment about additional detail needed to properly evaluate the study design. More detailed comments below.

Validity of the findings

See above comment about statistical details. I could not see the underlying data in the materials I received. More detailed comments below.

Additional comments

Cerrillo-Espinosa et al. present a metabarcoding study describing biodiversity patterns across different depths and regions of the Gulf of California.
The manuscript benefits from a nice dataset and uses it to assess an interesting theoretical question (the deep refuge hypothesis) related to potential community resilience in the face of environmental change and anthropogenic impact. Nevertheless, we have a few concerns that should be addressed before publication.
1. It is difficult to fully evaluate the manuscript as there is some ambiguity with regards to the methods. For example:
• Were all samples associated with the bottom at respective stations or were some/all pelagic in nature?
• Were diel sampling times comparable across sites? If not this should be identified as a potential limitation.
• Were deep water samples merged with mid-depth samples for some comparisons (e.g. Fig 3)?
• How were triplicate samples handled statistically? This is a notable concern since some of the statistics used (e.g. ANOSIM) don’t handle random effects well.
• What was the rationale for using ANOSIM in addition to PERMANOVA when the latter could presumably be used for all the multivariate statistics?
2. While eDNA metabarcoding is increasingly becoming more accepted as a biodiversity survey tool, there remain many skeptics based on perceived limitations. It is therefore important to address potential concerns about eDNA fate (persistence and drift) and potential biases (e.g. related to marker choice). We noted some brief coverage in Lines 478-486 but this could be enhanced. You will see some relevant references on this topic in Cote et al. 2024 (a paper addressing similar questions as yours that is already in your literature cited).
3. The storyline in this paper could improve. It is not well articulated what the novel findings are beyond the regional scale. Vertical separation of metabarcoded communities has been identified in a few studies, as has relatively high levels of diversity at the “deep” layers sampled in this study (as referenced in the MS and in several other studies using conventional methods). But perhaps the authors have specific reasons why they might not have expected these findings in their study area. Interpreting the data in the context of the deep refugia hypothesis was an interesting and original angle to apply this regional dataset but the discourse is spread across a few paragraphs. For example, the paragraph starting at line 441 is directly relevant but does not mention refugia until near the end. Similarly, topics like MPAs that are featured in the discussion are not mentioned in the Introduction. A more concerted effort on the flow of the discussion and the topics within would help the presentation of this work.
In closing, we feel this manuscript would contribute to the current state of knowledge once the above issues are addressed. More detailed comments are provided below. We wish the authors best of luck with their submission.
Sincerely,
David Coté and Gordon de Jong*

*Gordon de Jong is a Ph.D. candidate studying metabarcoding. He contributed to this review as part of his training. Both David and Gordon have reviewed the entire document and approved the comments.

Detailed comments:
Line 36: “deep ecosystems (40-500m)” this implies that SCUBA was done at 500m too?
Line 41: It would be useful to quickly summarize what the DWRH is
Lines 63-65: Is there a reference stating that this is mid-layer? It is of course relative but for many readers these depths would be considered epipelagic.
Line 72: Suggest removing "in them" as the structure of the sentence makes the reference to "them" ambiguous. There are also redundancies between presence and distribution. More concise to say, "...knowing their distribution in time and space."
Lines 74-75: In several regions commercial fish monitoring means knowledge of mid layers is better than shallow. E.g. North Atlantic.
Line 76: typo on constraints
Lines 78-79: Trawls and other fisheries data are widespread at mid depths. Maybe distinguish with the fact that your methods are non-destructive.
Line 79: change "that could" to "that has"
Line 81: Suggest removing “taken remotely”. It does not have to be taken remotely. Many studies actively take eDNA samples, especially in coastal environments.
Line 83: Would suggest changing “metabarcode” to “barcode”
Line 86: Suggest changing “mid-water depths, and deep-water” to "mid-water and deep-water depths"
Line 89: Typo on “refuge”.
Line 92: This last clause turns this sentence into a run-on sentence. Consider splitting. Alternatively we suggest adding this clause to the sentence that refers to refuge, as those ideas are linked better.
Lines 93-100: From here to the rest of the paragraph you cite literature that seems to answer your key questions - just not with eDNA methods. You may want to articulate more clearly that those methods may have biases that eDNA may be able to overcome and / or complement (as lightly alluded to in the next paragraph).
Line 100: this would be a good place to identify the information gap that your study was designed to fill.
Line 101: please provide reference for "globally recognized hotspot"
Lines 104-107: “…more than half total Mexico’s marine fisheries” suggest changing to “….more than half of Mexico’s marine fisheries”
Line 111: Suggest adding “within the Gulf of California” after “ecosystems” to make the statement more specific. Also please provide a reference for this statement.
Line 115: Confused about the “paired water samples”? Were SCUBA and Niskins used to collect eDNA from each depth? Please provide clarity.
Line 116: Should this be shallow, mid and deep water ecosystems? Because you have stated in your methods that you sampled shallow (<30m), mid-water (30 to 150m) and deep water (>150m)
Line 117: Suggest removing “of” between “metabarcoding” and “a fragment”
Lines 118-120: were these reefs sampled?
Line 120: Suggest changing “troubled” to “turbid”
Line 121: Suggest changing “like” to “such as”
Lines 121-124: Both goals are similar. Both state how communities, whether stated as eukaryotes or communities, change as a function of depth. Suggest making the goals specific and more distinct from one another.
Line 134: Regarding “sea water”. You use both "seawater" and "sea water" within the manuscript. We suggest keeping it consistent throughout the entire paper. In most cases it could be simplified to “water”.
Line 137: We suggest removing the first “the”
Line 140: CCW doesn't have the depth zone indicated like the rest
Line 144: Add “the” between “and” and “University”
Line 146: These characterizations of mid and deep could be misleading so it would be worth adding a reference to why they are called that.
Line 147: “live”? What do you mean by this?
Lines 149-150: Redundancy with the previous paragraph. Also, were the mid and deep samples collected close to the sea floor? Was there a consistent depth from the bottom these samples were collected?
Lines 156-159: Were any precautions taken in the field to minimize contamination during transfer from the sample container to filtering? Also, were the Niskin and Nalgene bottles decontaminated between uses?
Lines 158-159: Given the potential for diel vertical migration, was time of day accounted for in sample collections? What about time of year?
Lines 160-161: This makes it sound like both deep and shallow sites were filtered through the same filter.
Lines 167-169: Where was the running freshwater collected from? Was this distilled water?
Line 168: We suggest replacing “on the field” with “in the field”
Line 172: Suggest removing “Total” as it is difficult to extract all DNA
Line 184: We believe “Partial sequence” is incorrectly phrased. You sequenced a part of the COI gene region, not partial sequences. Suggest removing “Partial”
Line 189: Was PCR2 library prep? were the adapters included within the primers used in PCR1, or did you add the Illumina adapters separately in PCR2? The wording is a little confusing. Also, did the adapters have their own PCR conditions?
Line 202: Were the triplicate field samples pooled before sequencing? Or were results combined afterwards or treated as true triplicates?
Line 209: Is “quantified pooled” supposed to be “quantified and pooled”?
Lines 211-212: What was the sequencing depth?
Lines 227-228: Which database did you use for your BLAST? NCBI's nucleotide database? Also, when did you access this database? Should add a date of access for whichever database was used.
Line 236: What about freshwater taxa? Were they also removed?
Line 237: Also recommend adding the date BOLD was accessed.
Line 150: Species richness? OTU richness? Please clarify.
Line 252: Might be useful to the reader to know that Wilcoxon test is the non-parametric equivalent of a paired t-test.
Line 255: nMDS doesn't test for differences but rather is used to visualize differences. ANOSIM, listed below, is used to test for differences.
Line 257: How did a 1 way ANOSIM address 3 variables?
Lines 257-261: What was the rationale for doing both ANOSIM and PERMANOVA? Couldn’t PERMANOVA be used for all the multivariate analyses?
Line 262: Please provide a reference for the jaccard test. Most will be familiar with the index. I would think the test should be PERMANOVA. Also, please clarify the comparisons being done here.
Lines 264-268: Did pairwise tests account for multiple testing errors? It might be worth considering if you need every one of these pairwise tests.
Line 273: Were the dependencies among triplicate samples accounted for? Also, Bray Curtis is another dissimilarity matrix/index.
Line 274: The richness of what? OTUs? Species?
Lines 277-298: Good paragraph!
Line 300: “diverse phyla” - do you mean abundant? The figure to refers to the number of OTUs and the relative abundance.
Line 302: “deep-sea water samples” - did you combine mid-water 30-150 m and deep water 150+ m samples? This confusion is propagated throughout the results and the discussion.
Line 317: “cumulative richness” – do you mean OTU or species richness?
Line 319: “alpha diversity” – do you mean alpha richness?
Lines 321-325- How were the triplicates handled?
Line 341: I don't understand what "less contrasting" means in this context. It looks like the mid layer MDS cloud is more separated from the shallow samples than the deep water MDS cloud.
Lines 343-344: The methods state the TSW extended to 130m and therefore overlapped between surface and middle water samples.
Line 344: Suggest removing “it”.
Line 346: Suggest changing "basically influence" to "influenced". Also, we don't know if this is causation or correlation. So better to use a word like “association”.
Line 348: Suggest changing “separate” to “separated”
Line 349: Suggest changing “water mass” to “across water masses”
Line 350: “high ordination” – not familiar with this term.
Lines 350-351: Why was this test done separately? Couldn't everything be done within PERMANOVA?
Line 352: Suggest rephrasing “global level” as it can cause confusion to the readers. Perhaps changing to: “The eukaryotic community structure based on the presence-absence of species was not statistically significant different (Global R statistic: XXX)”

Line 353: Suggest changing “difference” to “differences”
Line 360: Suggest replacing "depth:" with "the"
Line 361: Not surprising given how few deep data points were available.
Line 362: Individual OTUs were not significantly different if you used presence - absence data, rather the communities of OTUs significantly differed. We suggest rewording accordingly.
Line 364: The "s" on OTUS shouldn't be capitalized.
Line 367: What is "species relative abundance richness"?
Line 372: Insert "were" before "classified"
Line 374: Which sea water masses?
Line 375: Suggest replacing “Not assigned” with “unassigned”.
Line 376: Add “the” before “Central”.
Line 384: suggest deleting “mean”.
Lines 384-385: There were only 3 deep-water samples, do you think that was a sufficient sample size to determine that this is always the case? Please make sure you acknowledge the limitation of such a small deep water sample size in your discussion.
Line 385: Suggest removing "s" from “OTUs” and suggest replacing "than" with "compared to".
Line 386: Remove “statistically”.
Line 388: What does substantial turnover of OTUs mean?
Line 391: Suggest changing “vertically depth gradient” to “vertical depth gradients”.
Lines 394-395: Consider changing the terminology to “general biodiversity primers”. There is no primer set that can detect all animals or groups.
Line 395: Suggest removing significant.
Line 398: If taxa are associated with water masses and the water masses change position, then deep water refugia could work for more species.
Line 405: Should combine the words “photo autotrophic”.
Line 406: Replace "seems" with "are".
Line 415: What do you mean by "collected remotely"? Adding some examples in parentheses would be useful if you are talking about specific technologies, such as AUV sampling.
Line 417: “lacked enough data” for what? We recommend replacing with “thus lacking data” or “data limited”.
Line 418: Remove "s" from OTUs.
Line 420: “principle of complementarity” should be explained.
Line 421: Do MPAs in the Gulf of California not protect the water column? If not, this should be highlighted. This also brings back the question of whether all your samples are pelagic in nature or were some/all associated with the bottom?
Lines 424-431: The example of copepods not migrating in circadian rhythm is not necessarily a good example of influencing genetic connectivity. Timescales of migration would have to be very long to support genetic divergence. Not sure of the relevance of this is anyway.
Line 427: change “…species that do not migrate of when migrate do not cross distinct vertical ecosystem” to “…species that do not migrate across distinct vertical ecosystem boundaries”
Line 428-429: Somewhere you need to address the possibility that species that are found in more than one habitat could be because of water mixing. There are studies contradicting that notion but its a cross that eDNA studies have to bear.
Lines 430-431: It would seem that a discussion of vertical migration should include reference to the Canals et al. eDNA study (already referenced in this MS).
Lines 432-435: Sentence should be reworded. The last clause doesn't fit well.
Line 437: Water column is referenced here but its not clear if we are talking about benthic species or not.
Line 438: We think this should read "was exclusive to".

Line 439: Not sure if this should be "exclusive of" or "excluded from".

Lines 441-444: Reword to avoid a run-on sentence.
Lines 441-458: Suggest revising this paragraph and relating more to the DWRH.
Line 447: remove "s" on “depths” and change “ecological” to “ecologically”.
Line 449: Unclear if the redistribution is away from or to these mid-water depth reefs. The wording of this sentence could also be improved.
Line 454: Suggest adding “the” between “within present”.
Lines 455-456: Is the reference to “data” yours or those from other studies?
Line 457: Suggest removing “observation”.
Line 457-458- What do you mean by “vertically stratified management”?
Line 460: In most cases in the MS "sea water" can be simplified to "water".

462: Should replace “with” to “the”.
Line 464: Suggest removing “a”.
Lines 468-473: eDNA results from sample to sample can have a lot of variation (Stat et al 2017, Cote et al. 2024), so it is hard to make inferences from a single sample. Also, due to the length of the sentence, it's hard to remember that the distinct zooplankton regions at the end of the sentence are linked to the eDNA community differences at the beginning.
Lines 476-486: It seems like a new paragraph should start here.
Line 478: Please provide references regarding “logistical challenges”. Also I don't think these are logistical challenges. They are more methodological in nature.
Line 483: Canals et al reference already cited could be used as evidence that eDNA is robust to vertical drifting since diel patterns were discernable.
Line 484: Should replace “in function” to “as a function”.
Lines 483-486: Deep ocean and shelf depths can also show issues of false negatives (see Cote et al. 2024 and He et al. 2023). Another issue with the deeper environments is the poorer taxonomic coverage of barcodes.
Lines 487-491: This is not a great closing paragraph. Would be good to end with some broad implications.
Lines 487: Should remove “the slopes of”.
Figure 1: It would be good to highlight the shapes of shallow and midwater depths. “Bathymetric” is spelled wrong. It’s confusing that the black line on the right panel corresponds to the gray line on the left panel, whereas the solid black lines on the left panel correspond to the red lines in the right panel (I think?). It's also very difficult to find the station names.
Figure 2: Is deep in this case mid + deep? If so, provide the rationale behind doing this.
Figure 3: Suggest to reevaluate the value of this plot. I would expect to see notable differences in community structure from 40m to 500m yet these are grouped together. Not sure it make sense to group them this way.
Figure 4: Couldn't these panels be combined, showing north (shallow and deep) next to central (shallow and deep)? That way, interactions could be assessed. Also, is it possible that the sites are more varied in the north than in the central area? Additionally, the whiskers of a box plot typically show 1.5 times the interquartile range, not 95% confidence limits. Please double-check if it is a 95% confidence interval.
Figure 5: As with everywhere else, it is difficult to understand what is deep. There are only 3 deep sites, but many more mid-depth sites. We suggest keeping it consistent throughout the paper and figures.
Figure 6: Please state that this plot is based on the presence/absence data. Also in plot a) is it worth including a single southern point?
Figure 7: The dependent variable should be the y axis.

---

## Round 0.2 · Minor Revisions

Dear Authors, the reviewer is satisfied with your response and the improvement of the manuscript. However, there are still very few typos that need to be taken into account.

Reviewer 1 ·

Basic reporting

I am satisfied that the authors' revisions have addressed my concerns from my initial review.

Experimental design

I am satisfied that the authors' revisions have addressed my concerns from my initial review.

Validity of the findings

I am satisfied that the authors' revisions have addressed my concerns from my initial review.

Additional comments

I am satisfied that the authors' revisions have addressed my concerns from my initial review.

I have a few very minor notes on the revised manuscript:

Line 135, should "> 30 m" actually be "< 30 m"?
Line 230: RedGel should be GelRed
Lines 422-431: were any of the indicator taxa discussed in this section detected in the eDNA? This should be mentioned either way.
Line 506: "considerably" should be "considerable"
Lines 540-543: This statement is true, but I think the broader result from the study is that the DRRH is *not* supported for the majority of taxa, especially given the statistically supported differences between the shallow and deep communities.

---

## Round 0.3 · accepted · Accept

Dear Authors, I am happy to inform you that your paper can be accepted for publication in PeerJ. The reviewer and I are fine with your last revision.